# Hydrogen Sulfide Metabolizing Enzymes in the Intestinal Mucosa in Pediatric and Adult Inflammatory Bowel Disease

**DOI:** 10.3390/antiox11112235

**Published:** 2022-11-12

**Authors:** Nathalie Stummer, Daniel Weghuber, René G. Feichtinger, Sara Huber, Johannes A. Mayr, Barbara Kofler, Daniel Neureiter, Eckhard Klieser, Sarah Hochmann, Wanda Lauth, Anna M. Schneider

**Affiliations:** 1Department of Pediatrics, Salzburger Landeskliniken (SALK) and Paracelsus Medical University (PMU), 5020 Salzburg, Austria; 2Research Program for Receptor Biochemistry and Tumor Metabolism, Department of Pediatrics, Salzburger Landeskliniken (SALK) and Paracelsus Medical University (PMU), 5020 Salzburg, Austria; 3Institute of Pathology, Salzburger Landeskliniken (SALK) and Paracelsus Medical University (PMU), 5020 Salzburg, Austria; 4Cell Therapy Institute, Spinal Cord Injury and Tissue Regeneration Center Salzburg (SCI-TReCS), Paracelsus Medical University (PMU), 5020 Salzburg, Austria; 5Department of Mathematics, Paris Lodron University, 5020 Salzburg, Austria

**Keywords:** hydrogen sulfide, H_2_S metabolism, inflammatory bowel disease, IBD pathogenesis, enzymatic expression

## Abstract

Hydrogen sulfide (H_2_S) is a toxic gas that has important regulatory functions. In the colon, H_2_S can be produced and detoxified endogenously. Both too little and too much H_2_S exposure are associated with inflammatory bowel disease (IBD), a chronic intestinal disease mainly classified as Crohn’s disease (CD) and ulcerative colitis (UC). As the pathogenesis of IBD remains elusive, this study’s aim was to investigate potential differences in the expression of H_2_S-metabolizing enzymes in normal aging and IBD. Intestinal mucosal biopsies of 25 adults and 22 children with IBD along with those of 26 healthy controls were stained immunohistochemically for cystathionine-γ-lyase (CSE), 3-mercapto-sulfurtransferase (3-MST), ethylmalonic encephalopathy 1 protein (ETHE1), sulfide:quinone oxidoreductase (SQOR) and thiosulfate sulfurtransferase (TST). Expression levels were calculated by multiplication of the staining intensity and percentage of positively stained cells. Healthy adults showed an overall trend towards lower expression of H_2_S-metabolizing enzymes than healthy children. Adults with IBD also tended to have lower expression compared to controls. A similar trend was seen in the enzyme expression of children with IBD compared to controls. These results indicate an age-related decrease in the expression of H_2_S-metabolizing enzymes and a dysfunctional H_2_S metabolism in IBD, which was less pronounced in children.

## 1. Introduction

Hydrogen sulfide (H_2_S) was long known only as a flammable, toxic gas with the smell of rotten eggs [1]. Like nitric oxide (NO) and carbon monoxide (CO), H_2_S now belongs to the family of gasotransmitters [2]. In contrast to the deleterious effects of H_2_S as an environmental pollutant, it has important regulatory functions in the body, which are assumed to vary depending on the cell type and concentration [3,4].

In human cells, endogenous H_2_S is the product of direct enzymatic desulfhydration of L-cysteine with or without homocysteine and indirect desulfhydration from 3-mercaptopyruvate (3MP), which is produced by cysteine aminotransferase (CAT) from cysteine and α-ketoglutarate (Figure 1) [5,6]. The former is catalyzed by cystathionine-γ-lyase (CSE) and cystathionine-β-synthase (CBS), whereas the latter is catalyzed by 3-mercapto-sulfurtransferase (3-MST) [5,7,8]. CBS can produce not only H_2_S and serine from cysteine but also H_2_S and lanthionine from two cysteine molecules and H_2_S and cystathionine from cysteine and homocysteine. Similarly, CSE catalyzes the conversion of cysteine into H_2_S, ammonia (NH3) and pyruvate as well as cysteine and homocysteine into H_2_S and cystathionine [9].

The enzymatic pathway, which involves CAT and 3-MST, converts cysteine and α-ketoglutarate into H_2_S and is found primarily in mitochondria [9,10]. CBS and CSE, however, reside in the cytosol but can translocate into mitochondria if 3-MST activity is reduced in conditions like hypoxia [10,11,12]. Additionally, endogenous H_2_S can be produced through D-amino acid oxidase (DAO). This pathway predominantly takes place in the cerebellum and the kidney and primarily involves the peroxisomes, where D-cysteine is converted into 3MP by DAO [13]. Then, 3MP is further processed to H_2_S by 3-MST in the mitochondria [12].

Likewise, the other H_2_S-producing enzymes are described as being organ-specific. CBS is largely expressed in the brain, the nervous system and the liver and CSE in the vascular system and liver, while 3-MST is mainly found in the brain and vascular system [14,15,16]. Nevertheless, the enzymes are found distributed across various tissues [11,15]. The main H_2_S-producing enzymes in the gastrointestinal (GI) tract seem to be CBS and CSE [17,18,19,20,21]. The CAT-3MST pathway, however, might play an even more important part in colonic H_2_S production [22].

The colonic epithelium is not only exposed to endogenously produced H_2_S but also to exogenously produced H_2_S on its luminal side [9]. This is the result of proteins being degraded by intestinal bacteria into amino acids, including cysteine and other sulfur-containing compounds [9,23]. The amount of sulfide in the colonic lumen not only depends on the composition of the microbiota but also on the dietary status, in particular the protein intake [24,25]. Some dietary compounds that reach the colon without being absorbed, like zinc, heme and polyphenols, are able to bind H_2_S [26,27,28].

Most of the H_2_S exits the body as sulfate via urine through the kidney. Sulfide:quinone oxidoreductase (SQOR), a flavoenzyme bound to the inner mitochondrial membrane, oxidizes H_2_S [29]. This process generates sulfane sulfur while transferring electrons to the coenzyme Q (CoQ) [30]. The transfer of sulfane sulfur to glutathione (GSH) and sulfite (SO_3_^2−^) by SQOR results in glutathione persulfide (GSSH) or thiosulfate (SSO_3_^2−^) [29,31,32]. Another mitochondrial matrix enzyme, ethylmalonic encephalopathy 1 protein (ETHE1, also known as persulfide dioxygenase), oxidizes GSSH and cuts the disulfide bond, producing SO_3_^2−^ and GSH [33,34,35]. Sulfite oxidase (SUOX), which is found in the mitochondrial intermembrane space, further oxidizes sulfite to sulfate (SO_4_^2−^) and transfers two electrons to cytochrome c [36].

Thiosulfate sulfurtransferase (TST), also known as rhodanese, represents another mitochondrial enzyme which is able to transfer sulfane sulfur from GSSH to thiosulfate [32,37,38]. Sulfide-detoxifying enzymes, like rhodanese, then catalyze the transformation of thiosulfate into sulfate or sulfite, which is immediately oxidized into sulfate [39,40].

The need for a quick removal of H_2_S from the body is partly due to its toxic effects on mitochondria. It inhibits complex IV (cytochrome c oxidase, COX), which is a component of the electron transport chain [41]. This bioenergetic modulation is a concentration-dependent effect, since in lower concentrations, H_2_S even enhances the electron transport chain during oxidative phosphorylation by acting as an electron donor to complex II [42,43,44]. H_2_S is the first inorganic compound known to be coupled with ATP generation in mitochondria [45]. Of all cells, the intestinal epithelial cells (IEC) use this function of H_2_S most efficiently [46,47].

H_2_S is obtaining more and more attention as a key player in various diseases [48,49,50,51,52,53,54,55,56,57,58]. As the colon is exposed not only to endogenously but also to exogenously produced H_2_S, it is exposed to a greater amount than any other organ. This is also the reason why H_2_S is receiving growing attention in intestinal diseases, especially in the unknown etiology of inflammatory bowel disease (IBD) [9,17,18,59,60,61,62].

IBD is the umbrella term for two chronic relapsing diseases, namely ulcerative colitis (UC) and Crohn’s disease (CD), which are defined by clinical, pathological, endoscopic and radiological characteristics of inflammation in the GI-tract [63]. CD shows transmural inflammatory “skip lesions” that can develop anywhere in the GI-tract but most commonly occur in the terminal ileum [64,65,66]. UC shows homogenous inflammatory lesions limited to the mucosa in the colon and mainly affects the distal rectum [65,66,67].

Globally, the incidence of IBD is increasing, especially in newly industrialized countries. This, together with the increasing rates of diagnosis and decreasing rates of mortality, has led to an enormous rise in the number of people affected by IBD [68]. These trends highlight the significance of elucidating the exact pathogenesis, which is still unknown, in order to be able to develop targeted therapies. This is especially important for pediatric IBD, as its progression is more aggressive and rapid and the intestinal involvement is more extensive when compared to adult IBD [69,70]. The current understanding of the pathogenesis of this disease indicates an interplay of genetics, external environment, intestinal microbiota and altered immune response [71,72,73,74].

As with many functions of H_2_S, its effect on intestinal inflammation varies at different concentrations [18]. Interestingly, H_2_S not only exhibits pro-inflammatory but also anti-inflammatory effects [17,19,75,76,77,78,79,80]. This leads to the assumption that H_2_S might have a dome-shaped curve of beneficial concentration levels in the colon, with levels which are too high and too low being harmful.

Some studies have already shown differences in the expression of selected enzymes involved in the metabolism of H_2_S in inflamed tissue and healthy controls [17,19,20,61,80,81,82]. To be able to examine the differences in IBD patients of different age groups, the physiological changes in the H_2_S metabolism through aging have yet to be assessed. This study aims to investigate differences in the expression of both producing (CSE, 3-MST) and detoxifying enzymes (ETHE1, SQOR and TST) in colon tissue samples, not only in pediatric and adult patients with IBD, but also throughout aging in comparison with healthy controls.

## 2. Materials and Methods

The study was approved by the local ethics committee (415-E/2080/5-2016) and conducted in accordance with the Helsinki Declaration of 1975 (revised 2013).

### 2.1. Patients and Tissue Samples

Tissue samples were taken during colonoscopies at the Departments for Pediatrics and Internal Medicine, University Hospital of Salzburg, Austria. These colonoscopies were performed on patients with abdominal symptoms in order to rule out or confirm IBD. For better representation of the inflammation status of the intestine, the terminal ileum and all segments of the colon (cecum, ascending colon, transverse colon, descending colon, sigmoid colon, and rectum) were biopsied in routine ileo-colonoscopies. The tissue samples were then analyzed to confirm or exclude the diagnosis. The remaining tissue samples, which were prepared by the Institute of Pathology, University Hospital Salzburg, were used for the subsequent immunohistochemical staining for this study. Patients’ characteristics are shown in Table 1.

The samples were classified according to a previous study by Schneider et al. [83]. Patients with macroscopically and histologically proven IBD were assigned to the study group. In this category, samples were taken from the terminal ileum in CD patients and from the rectum of UC patients, since these represent the most frequently affected locations. To exclude any treatment-based effects on the colonic epithelium, only newly diagnosed patients were selected for this study. An additional sample was obtained from the ascending colon in the whole study group. This provided a better comparison of the different categories by analysis of the same colon segment.

If the results of the colonoscopy excluded UC and CD, they were appointed as controls. The analysis was conducted by the pathologists using a histologic severity score (HSS) in accordance with the study by Schneider et al. [83]. Zero to three points were given in each of the categories of crypt architecture, signs of acute inflammation, signs of chronic inflammation and regeneration of the epithelium, resulting in a score range of 0–12. Subsequently, the mean score of these categories was calculated. To qualify for the control group, the samples had to have a value of 1 or less. Analogous to the study group, samples from the terminal ileum, rectum and ascending colon were analyzed in the controls.

The following groups are the result of this classification:-P-IBD: pediatric patients (below 18 years) with IBD-A-IBD: adult patients with IBD-Pediatric controls: children and adolescents (below 18 years) without signs of IBD or inflammation-Adult controls: adults without signs of IBD or inflammation

### 2.2. Immunohistochemistry

Formalin-fixed paraffin-embedded (FFPE) tissues were cut into 4 μm sections. The immunohistochemical staining was performed as described by Zimmermann et al. [84]. The following primary antibodies were used: SQOR (17256-1-AP, dilution 1:1500; Proteintech, Illinois, USA), 3-MST (NBP1-87147, dilution 1:1000; NovusBio, Colorado, USA), CSE (12217-1-AP, dilution 1:1000; Proteintech, Colorado, United States), TST (NBP1-87147, dilution 1:2000; NovusBio, Colorado, USA), ETHE1 (NBP1-81689, dilution 1:2000; NovusBio, Colorado, USA). These antibodies were diluted in Dako antibody diluent with background-reducing components (Dako, Glostrup, Denmark).

After deparaffinization, the samples were placed in TE-T buffer (1mM EDTA, 0.05% Tween-20, pH 8.0) for 40 min at 95 °C. Then they were washed with distilled H_2_O and equilibrated in a phosphate-buffered saline containing 0.5% Tween 20 (PBS-T; pH 7.6), followed by incubation in Dako Peroxidase Blocking reagent for 5 min and then, again, rinsed in phosphate-buffered saline and Tween 20 (PBS-T). Samples were incubated with the respective primary antibody for one hour at room temperature (RT) and rinsed in PBS-T. Afterwards, the second antibody (Dako EnVision+; Dako, Glostrup, Denmark) was added for one hour at RT and rinsed in PBS-T. The specimens were then incubated in diaminobenzidine for 5 min and subsequently counterstained with hematoxylin. Samples were blued for 10 min in running tap water, dehydrated in 2-propanol and xylol, and ultimately mounted on slides [84].

### 2.3. Evaluation and Statistics

After immunohistochemical staining, slides were scanned automatically at 20x magnification using the VS-120-L Olympus slide scanner 100-W system and processed using the Olympus VS-ASW-L100 program (Olympus, Shinjuku, Tokyo, Japan). The staining was evaluated in accordance with a previous study by Schneider et al. [83]. First, the staining intensity of IEC was assessed using a score system ranging from 0 to 3 (0 = no staining, 1 = weak staining, 2 = moderate staining, 3 = strong staining) (Figure 2). Then, the percentage of positive IEC inside the crypts was estimated. This was conducted by two independent examiners blinded to each other. The mean values were calculated, and the mean staining score was multiplied with the mean percentage of stained cells. These values were referred to as expression levels [84]. Biopsies without IEC were excluded.

The results were illustrated with boxplots. For statistical analysis, R-Studio (R version 4.1.1, Vienna, Austria) was used. Owing to the ordinal scale of the expression scores, data were compared by median and interquartile range. Then, the nonparametric ANOVA-Type statistic was applied with rankFD. Additionally, the relative effect (pd) was calculated. It was tested for the null hypothesis in regard to the relative effect. A 5% significance level was used on all tests; therefore, a *p*-value below 0.05 indicated a significant difference. Since the analysis was of an exploratory nature, no *p*-value correction for multiple testing was applied.

## 3. Results

### 3.1. Age-Related Difference

The control group was examined in order to determine a possible difference between adults and children independent of any disease-related effect. Overall, a lower expression of most H_2_S-metabolizing enzymes was observed in adults compared to children in different parts of the gastrointestinal tract (Table 2, Table 3 and Table 4; Figure 3, Figure 4, Figure 5 and Figure 6).

In the terminal ileum, the expression of CSE was 62% lower in adults compared to children (*p* = 0.0086). The other enzymes also tended to be less expressed in the adult terminal ileum (reduction: 3-MST: 28%; ETHE1: 36%; SQOR: 28%; TST: 19%) (Figure 3; Table 2). Apart from SQOR, the expression of CSE, 3-MST, ETHE1 and TST in the rectum tended to be reduced by 9% to 37% in adults compared to children. (Figure 4; Table 3). In the ascending colon, the expression of MST was 42% lower in adults compared to children (*p* = 0.0224). A similar trend toward lower expression levels of these enzymes was also observed in the adult ascending colon (reduction: CSE: 21%, ETHE1: 21%; SQOR: 27%; TST: 19%) (Figure 5; Table 4).

### 3.2. Disease Related Difference

To determine whether there is a difference in the expression of H_2_S-metabolizing enzymes between the two IBD entities, the terminal ileum was examined for CD and the rectum for UC, as these sites are the most likely to be affected. For a general comparison with healthy controls, the ascending colon was examined. Since age-related differences were observed in healthy colon samples, the comparison was performed individually for adult and pediatric samples (Figure 6 and Figure 7).

### 3.3. Terminal Ileum (Crohn’s Disease)

Except for SQOR, the expression of H_2_S-producing and detoxifying enzymes in the terminal ileum of adult CD patients tended to be lower compared to healthy controls (reduction: 3-MST: 46%; CSE: 17%; ETHE1: 12%; TST: 34%) (Figure 8 and Table 5). In the terminal ileum of pediatric CD patients, the expression of CSE was 56% lower than in healthy children (*p* = 0.0396). The expression of 3-MST and TST also tended to be lower in pediatric CD patients, with a reduction of 7% and 22%, respectively. In contrast, the expression of ETHE1 and SQOR was similar in patients and healthy controls.

### 3.4. Rectum (Ulcerative Colitis)

The expressions of CSE, ETHE1, SQOR and TST were significantly lower in the rectum of adult UC patients compared to healthy controls (CSE: *p* < 0.001, reduction: 92%; ETHE1: *p* = 0.0017, reduction: 39%; SQOR: *p* = 0.021, reduction 50%; TST: *p* = 0.0101, reduction 41%) (Figure 9 and Table 6). There was also a downward trend in the expression of 3-MST (62% decrease).

In pediatric samples of UC patients, the expression of CSE was 62% lower compared to healthy controls (*p* = 0.0162). The levels of the other enzymes also tended to be lower in the rectum of UC patients compared to healthy controls (reduction: 3-MST: 13%; ETHE1: 5%; SQOR: 40%; TST: 18%).

### 3.5. Ascending Colon

The expression of H_2_S-producing enzymes 3-MST and CSE as well as of the detoxifying ETHE1 was 47% to 85% lower in samples of the ascending colon of adult CD patients (3-MST: *p* = 0.0188, reduction: 65%; CSE: *p* = 0.0004, reduction: 85%; ETHE1: *p* = 0.0026, reduction: 47%) (Figure 10, Table 7 and Table 8). The expression of SQOR and TST also tended to be lower in adult CD patients (reduction: SQOR: 34%; TST: 31%).

In adult UC patients, a similar regulation was observed. CSE and ETHE1 showed a 70% and 47% lower expression, respectively, compared to healthy controls (CSE: *p* = 0.0019; ETHE1: *p* = 0.0018). The expression of 3-MST, SQOR and TST tended to be lower as well in adult UC patients compared to healthy controls (reduction: 3-MST: 40%; SQOR: 23%; TST: 34%).

In contrast, in the ascending colon of pediatric CD patients, TST expression was significantly lower than in healthy controls (*p* = 0.0308, reduction: 10%). There was also a trend towards lower expression of 3-MST, ETHE1 and SQOR (reduction: 3-MST: 15%; ETHE1: 13%; SQOR: 10%).

In pediatric UC patients, the expression of SQOR and TST in the ascending colon was 42% and 22% lower than in healthy controls (SQOR: *p* = 0.0078; TST: *p* = 0.0118). 3-MST and ETHE1 expression also tended to be lower (reduction: 3-MST: 6%; ETHE1: 9%).

No difference was seen in the expression of CSE either in pediatric CD or in pediatric UC patients.

### 3.6. Correlation with Severity of Inflammation

To investigate whether or not the lower expression of H_2_S-metabolizing enzymes correlates with the degree of inflammation, the expression score and the HSS were subjected to a non-parametric Spearman correlation analysis, which evaluates monotonic correlations. This was conducted separately for pediatric and adult groups in the terminal ileum and the ascending colon for the CD group and the rectum and ascending colon for the UC group. No strong correlations were seen in either group (Appendix A: Figure A1, Figure A2 and Figure A3).

## 4. Discussion

In the present study, the expression of H_2_S-metabolizing enzymes was analyzed in colonic mucosal biopsies of (i) healthy children and adults as well as (ii) pediatric and adult patients with IBD in comparison to healthy controls.

(i)Our data suggest that the expression of H_2_S-producing and H_2_S-detoxifying enzymes decreases with age. This seems to be a result of the physiological process of aging, as a decrease of H_2_S-metabolizing enzymes was shown in various other organs during aging [85,86,87,88]. To our knowledge, there has been no study so far on age-related changes in the expression of H_2_S-metabolizing enzymes in the colon other than TST. Yi et al. reported a decrease of TST in the human colonic epithelial cells of 25- to 65-year-old males throughout aging by proteomic analysis [89]. Many studies confirm an age-related decline in the expression of certain metabolism-related proteins like the oxidative phosphorylation complexes in mitochondria in the intestine and other organs [90,91,92,93,94,95]. Özsoy et al. demonstrated that there is a continuous decline in the number of colonic crypts with a partial or complete loss of complexes I and IV [90]. On a cellular level, H_2_S affects nearly all “hallmarks of aging”, which are genomic instability, epigenetic alterations, loss of proteostasis, deregulated nutrient sensing, mitochondrial dysfunction, cellular senescence, stem cell exhaustion, altered intracellular communication and telomere attrition [96]. This suggests that H_2_S might play a modulating role in aging, even though its exact contribution to this complex process is still being debated. In colonocytes, H_2_S might exert genotoxic effects, as it does in human lung fibroblasts [97,98]. There, NaHS, a H_2_S donor, is seen to induce fibroblast cell death by triggering the release of cytochrome c and the translocation of Bax into the mitochondria. It also induces p53 and is responsible for cell cycle changes and micronuclei formation, which show an increased amount of DNA lesions left unrepaired [98]. This genotoxicity of H_2_S, just like many of its other functions, seems to be concentration dependent [97,98]. High concentrations inside the colon might lead to genomic instability and the accumulation of mutations, which is seen in colorectal cancer [97].(ii)Regarding IBD, independent of the disease subtype, diseased adults show a downward trend of lower expression of H_2_S-metabolizing enzymes compared to healthy controls.

As with aging, H_2_S has been reported to have positive and negative effects on intestinal disorders and inflammation, which is why its role is quite controversial [18,60]. Several studies on the changes in H_2_S-metabolizing enzymes associated with IBD have already been published [17,19,20,61,80,81]. The studies examining CSE are contradictory. Some results indicated a lower expression using Western blot analysis, while for example, Hirata et al. showed an increase in the enzyme expression using Western blot analysis [20,80]. Wallace et al. also reported an increase in CSE levels using immunohistochemical staining [17]. Yet, these studies were conducted in rodents, and the immunohistochemical staining showed an increase in CSE expression in the mucosa and submucosa while the epithelial cells remained unstained [17,80]. Consequently, this could still be in line with our results, which focused on epithelial cells and detected a lower expression in patients with IBD.

Studies on the other enzymes, namely ETHE1, TST and SQOR, all show a lower expression in human and in animal colitis samples [20,61,81]. These studies, together with our results, strengthen the assumption that the metabolism of H_2_S appears to be dysfunctional in IBD in adults. Whether this is the consequence of the disease or part of its cause is yet to be answered. The fact that children show a less pronounced reduction when compared to controls indicates that the reduction might be a cumulative effect of the inflammatory process as a consequence of the disease over time. Adults with IBD would thus exhibit a greater difference in the expression of H_2_S-metabolizing enzymes compared to healthy controls.

However, this is in contrast to the fact that lower expression scores do not correlate with the severity of the inflammation in our study, yet, the cross-sectional design of our study reflects the degree of inflammation at a single point in time rather than the actual cumulative inflammatory burden.

The lower expression presented here, together with other studies that show lower enzyme expression, supports the hypothesis that the expression of H_2_S-producing as well as H_2_S-detoxifying enzymes might in fact be lower in IBD patients [17,19,20]. The lower expression of enzymes of the H_2_S metabolism might lead to a loss of control of the H_2_S levels the colon is exposed to. Consequently, it could be more vulnerable to high or low levels of H_2_S in the lumen, both of which promote inflammation [18]. Lower enzyme expression of detoxifying enzymes leading to an increased level of H_2_S would thus predispose to the pro-inflammatory effects of H_2_S [17,75]. On the other hand, decreased levels of H_2_S due to the lower expression of H_2_S-producing enzymes might increase the inflammatory response caused by the lack of anti-inflammatory effects [19,76,77,99]. Since there are other plausible factors involved, like diet and the composition of the microbiota, which affect the H_2_S levels, it would be necessary to directly measure H_2_S levels in the colon. The increased susceptibility to inflammation because of the inability to control luminal H_2_S levels, however, could be the common denominator in many of the different hypotheses about the pathogenesis of IBD. Several studies suggest an alteration in H_2_S-producing microbiota as a decisive aspect in the development of IBD [61,100,101,102]. An increased level of H_2_S in the colonic lumen has been shown to destabilize the protective mucosal layer, which acts as a barrier between microbiota and the epithelium. This destabilization through reduction of disulfide bonds most likely increases the interaction of bacteria with the colonic epithelial cells [103]. Lower expression of detoxifying enzymes, as demonstrated in this study, may accelerate or even initiate this process by allowing an unopposed bacterial production of H_2_S. On the contrary, a lower expression of H_2_S-producing enzymes may compromise the intestinal repair mechanisms [17,80]. Furthermore, endogenously produced H_2_S has been shown to increase mucus production, inhibit the activation of the inflammasome pathway and protect against barrier dysfunction [62,104,105]. A decrease in these effects might contribute to the development or deterioration of IBD. As a result, no matter which way the H_2_S level scale is tipped, the body will be unable to restore balance. Nevertheless, further studies on changes in H_2_S levels inside the colon of patients with IBD are needed to elucidate the consequences of low expression of H_2_S-metabolizing enzymes.

Data on H_2_S-metabolizing enzymes in human samples are very limited, especially in children. To our knowledge, there is only one study, by Mottawea et al., which reported a lower expression of TST, SQOR and ETHE1 in pediatric CD patients by employing proteomic analysis of mucosal biopsies [61]. This is in agreement with our results showing a lower expression of most H_2_S-metabolizing enzymes, which is less extensive than the differences in adults. All, except for CSE in the ascending colon of children with IBD and ETHE1 as well as SQOR in the terminal ileum of pediatric CD patients, show a downward trend. Mottawea et al. propose that the lower levels are due to the lack of butyrate-induced gene expression [61]. However, the fact that the differences in enzyme expression in children are not as pronounced as compared to those in adults with IBD suggests that other factors might be necessary for the development of early-onset IBD.

Due to the limited sample size in each group, the statistical correlations are moderate. Even though the study does not take the exogenous H_2_S metabolism into account, it is the first study that comprehensively assesses the endogenous metabolism by not only analyzing the production but also the detoxification pathways. Another main strength of this study is the broad age spectrum of patients. Samples from patients over an age-span of 80 years offer the chance to elucidate the changes in the expression of these complexes throughout aging, which allows a better understanding of their role in the pathogenesis in different age groups.

## 5. Conclusions

A reduction in the expression of H_2_S-synthesizing and H_2_S-detoxifying enzymes, namely CSE, 3-MST, ETHE1, SQOR and TST, was observed in the comparison of colon samples from healthy children and adults as well as in the comparison of adult IBD patients and healthy controls. These results indicate a physiological decrease in the expression of H_2_S-metabolizing enzymes with aging and a pronounced dysfunctional H_2_S metabolism in adult patients with IBD compared to healthy controls. Unlike adult IBD patients, children with IBD show fewer differences from their control group. This suggests that the disruption of H_2_S metabolism is a consequence rather than a cause of inflammation in IBD, although this is limited by the cross-sectional nature of the study. However, more research is needed to answer this question and hopefully identify new targets for an effective treatment of IBD.

## Figures and Tables

**Figure 1 antioxidants-11-02235-f001:**
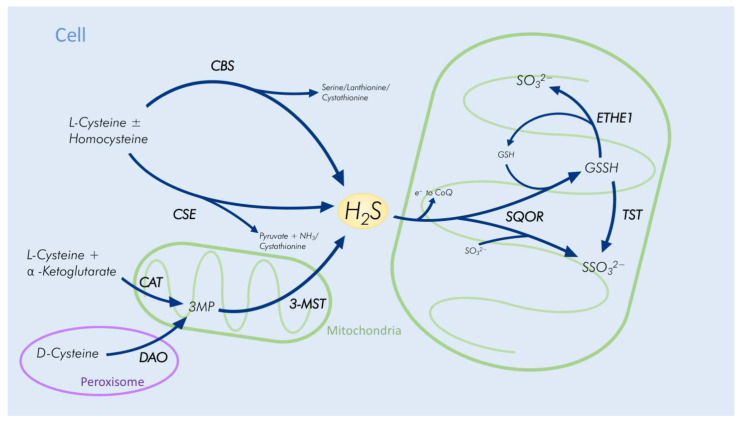
Endogenous hydrogen sulfide (H_2_S) production and detoxification. H_2_S is produced from L-cysteine with or without homocysteine by cystathionine-γ-lyase (CSE) and cystathionine-β-synthase (CBS) in the cytosol and from 3-mercaptopyruvate (3MP) by 3-mercapto-sulfurtransferase (3-MST) in mitochondria. The detoxification is catalyzed by sulfide:quinone oxidoreductase (SQOR) and subsequently by thiosulfate sulfurtransferase (TST) or ethylmalonic encephalopathy 1 protein (ETHE1). This process takes place in the mitochondria. Cysteine aminotransferase (CAT); D-amino acid oxidase (DAO); electron (e^−^), coenzyme Q (CoQ); glutathione persulfide (GSSH); thiosulfate (SSO_3_^2−^); sulfite (SO_3_^2−^); glutathione (GSH); ammonia (NH_3_).

**Figure 2 antioxidants-11-02235-f002:**
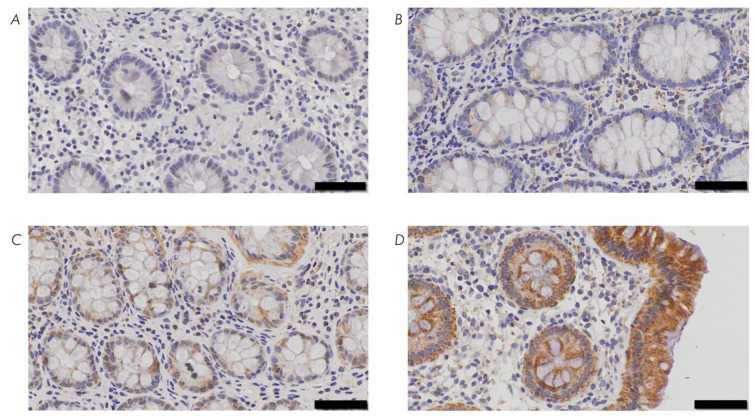
Assessment of staining intensity. (**A**) 0 = no staining (case number 36677: adult with Crohn’s disease; biopsy of the ascending colon stained with cystathionine-γ-lyase (CSE) antibody), (**B**) 1 = weak staining (case number 21480: child with Crohn’s disease; biopsy of the ascending colon stained with cystathionine-γ-lyase (CSE) antibody), (**C**) 2 = moderate staining (case number 10145: adult from control group; biopsy of the ascending colon stained with 3-mercapto-sulfurtransferase (3-MST) antibody), (**D**) 3 = strong staining (case number 13366: child from control group; biopsy of the ascending colon stained with ethylmalonic encephalopathy 1 protein (ETHE1) antibody). Scale bar: 50 μm.

**Figure 3 antioxidants-11-02235-f003:**
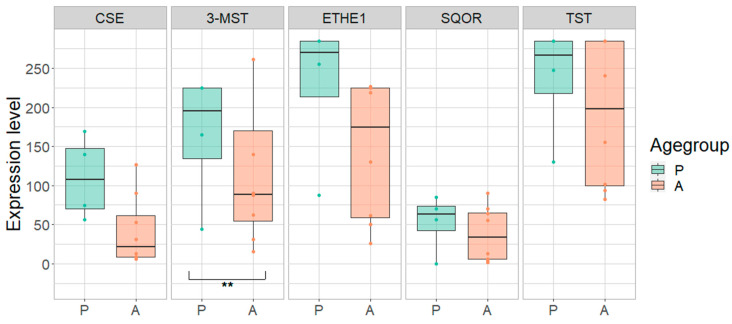
Expression of cystathionine-γ-lyase (CSE), 3-mercapto-sulfurtransferase (3-MST), ethylmalonic encephalopathy 1 protein (ETHE1), sulfide:quinone oxidoreductase (SQOR) and thiosulfate sulfurtransferase (TST) in pediatric (P) and adult (A) samples in the terminal ileum of pediatric (P) and adult (A) healthy controls. Range of expression levels from a minimum of 0 to a maximum of 300. ** = *p*-value < 0.01.

**Figure 4 antioxidants-11-02235-f004:**
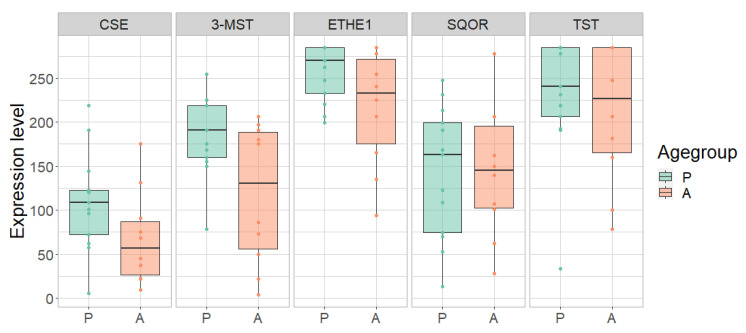
Expression of cystathionine-γ-lyase (CSE), 3-mercapto-sulfurtransferase (3-MST), ethylmalonic encephalopathy 1 protein (ETHE1), sulfide:quinone oxidoreductase (SQOR) and thiosulfate sulfurtransferase (TST) in pediatric (P) and adult (A) samples in the rectum of pediatric (P) and adult (A) healthy controls. Range of expression levels from a minimum of 0 to a maximum of 300.

**Figure 5 antioxidants-11-02235-f005:**
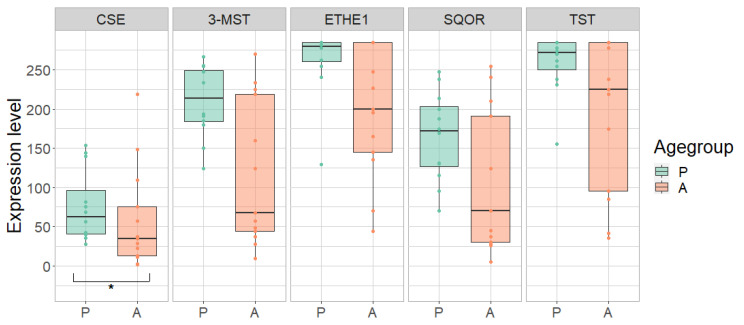
Expression of cystathionine-γ-lyase (CSE), 3-mercapto-sulfurtransferase (3-MST), ethylmalonic encephalopathy 1 protein (ETHE1), sulfide:quinone oxidoreductase (SQOR) and thiosulfate sulfurtransferase (TST) in pediatric (P) and adult (A) samples in the ascending colon of pediatric (P) and adult (A) healthy controls. Range of expression levels from a minimum of 0 to a maximum of 300. * = *p*-value < 0.05.

**Figure 6 antioxidants-11-02235-f006:**
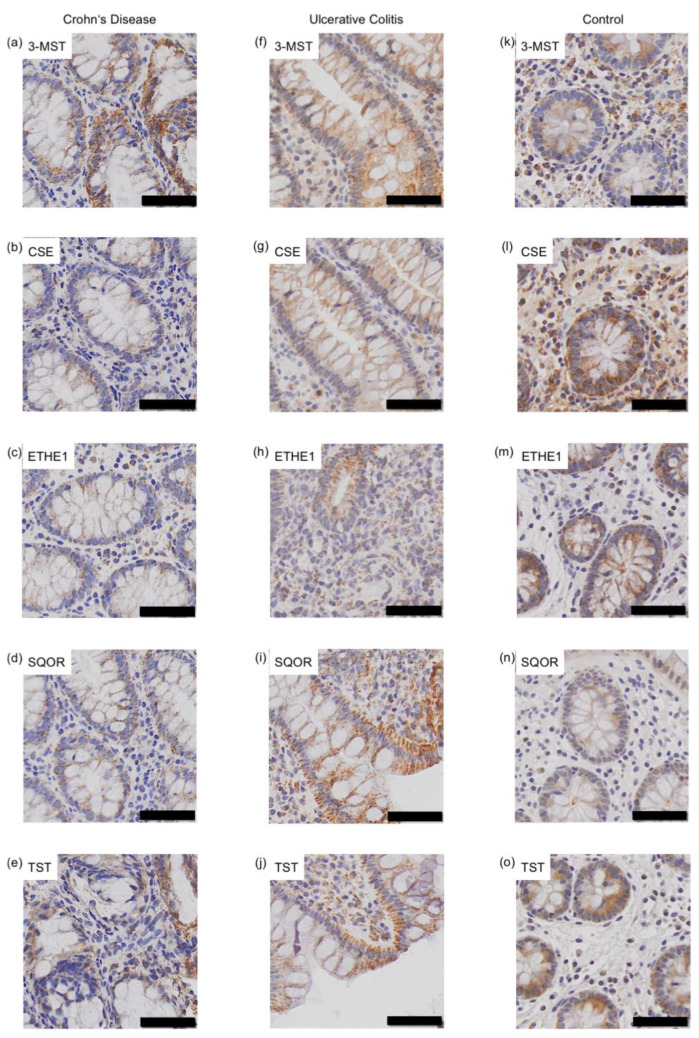
Immunohistochemical staining of 3-mercapto-sulfurtransferase (3-MST) (**a**,**f**,**k**), cystathionine-γ-lyase (CSE) (**b**,**g**,**l**), ethylmalonic encephalopathy 1 protein (ETHE1) (**c**,**h**,**m**), sulfide:quinone oxidoreductase (SQOR) (**d**,**i**,**n**) and thiosulfate sulfurtransferase (TST) (**e**,**j**,**o**) in the ascending colon of an pediatric Crohn’s disease (CD) patient (case number: 21480; **a**–**e**), ulcerative colitis (UC) patient (case number: 17921; **f**–**j**) and a healthy control (case number: 41952; **k**–**o**). Scale bar: 50 μm.

**Figure 7 antioxidants-11-02235-f007:**
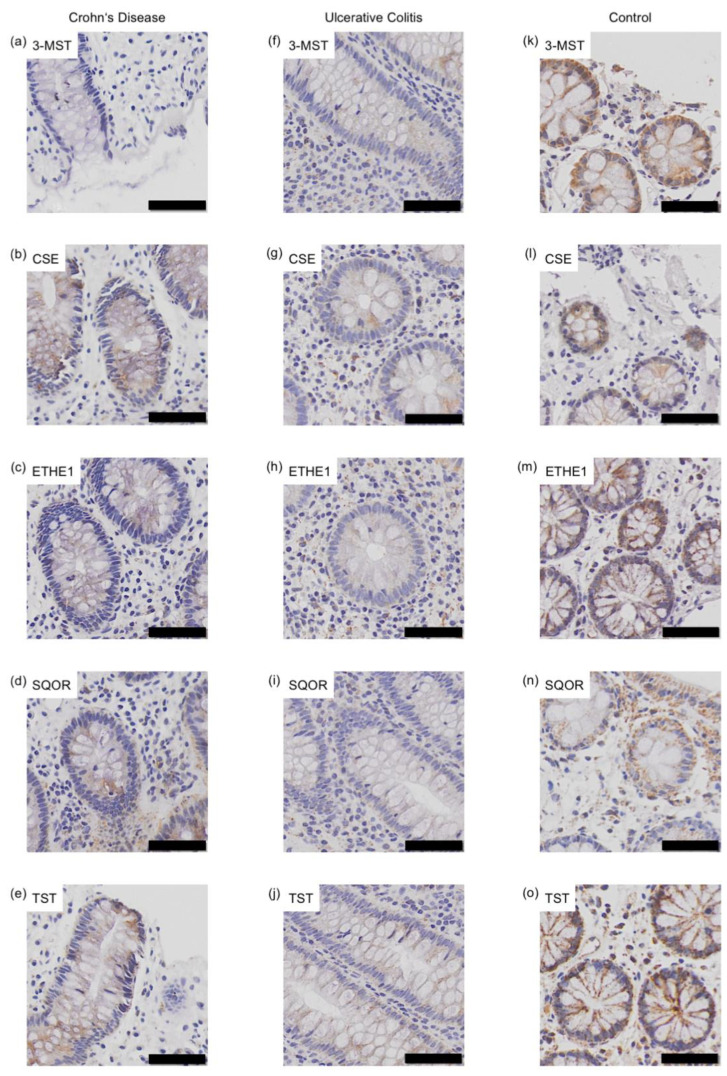
Immunohistochemical staining of 3-mercapto-sulfurtransferase (3-MST) (**a**,**f**,**k**), cystathionine-γ-lyase (CSE) (**b**,**g**,**l**), ethylmalonic encephalopathy 1 protein (ETHE1) (**c**,**h**,**m**), sulfide:quinone oxidoreductase (SQOR) (**d**,**i**,**n**) and thiosulfate sulfurtransferase (TST) (**e**,**j**,**o**) in the ascending colon of an adult Crohn’s disease (CD) patient (case number: 1593; **a**–**e**), ulcerative colitis (UC) patient (case number: 118884; **f**–**j**) and a healthy control (case number: 14411; **k**–**o**). Scale bar: 50 μm.

**Figure 8 antioxidants-11-02235-f008:**
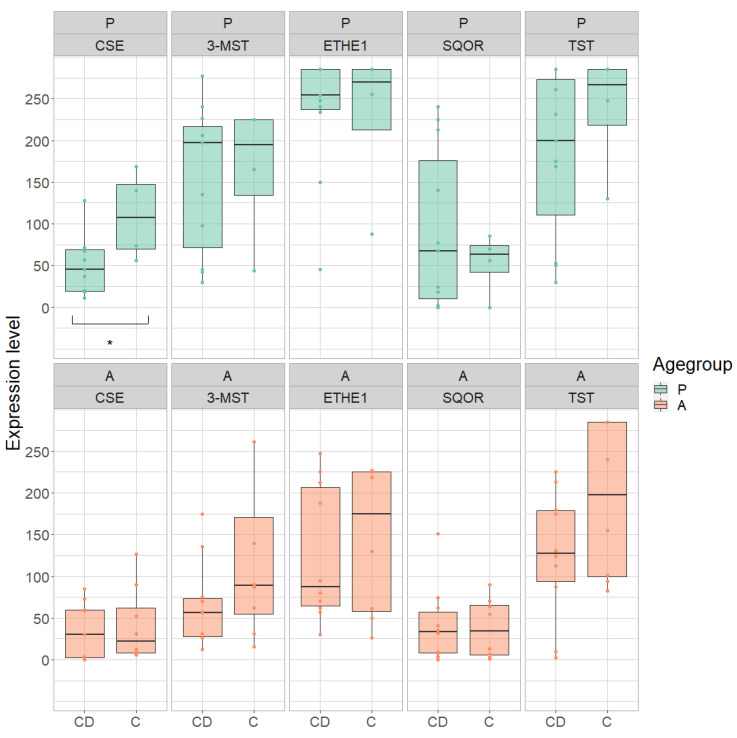
Expression of cystathionine-γ-lyase (CSE), 3-mercapto-sulfurtransferase (3-MST), ethylmalonic encephalopathy 1 protein (ETHE1), sulfide:quinone oxidoreductase (SQOR) and thiosulfate sulfurtransferase (TST) in pediatric (P) and adult (A) samples in the terminal ileum in Crohn’s disease (CD) compared to healthy controls (C). Range of expression levels from a minimum of 0 to a maximum of 300. * = *p*-value < 0.05.

**Figure 9 antioxidants-11-02235-f009:**
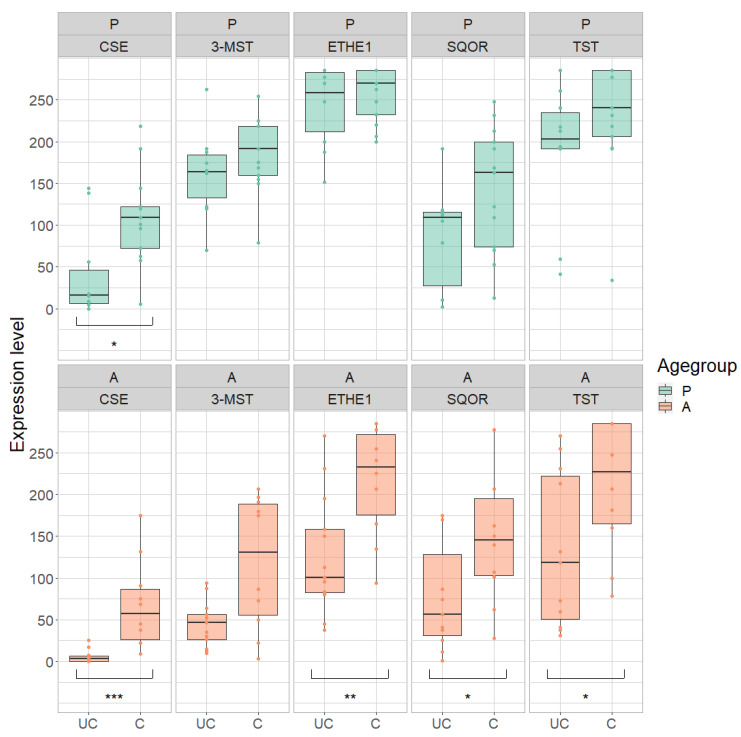
Expression of cystathionine-γ-lyase (CSE), 3-mercapto-sulfurtransferase (3-MST), ethylmalonic encephalopathy 1 protein (ETHE1), sulfide:quinone oxidoreductase (SQOR) and thiosulfate sulfurtransferase (TST) in pediatric (P) and adult (A) samples in the rectum of ulcerative colitis (UC) compared to healthy controls (C). Range of expression levels from a minimum of 0 to a maximum of 300. * = *p*-value < 0.05, ** = *p*-value < 0.01, *** = *p*-value < 0.001.

**Figure 10 antioxidants-11-02235-f010:**
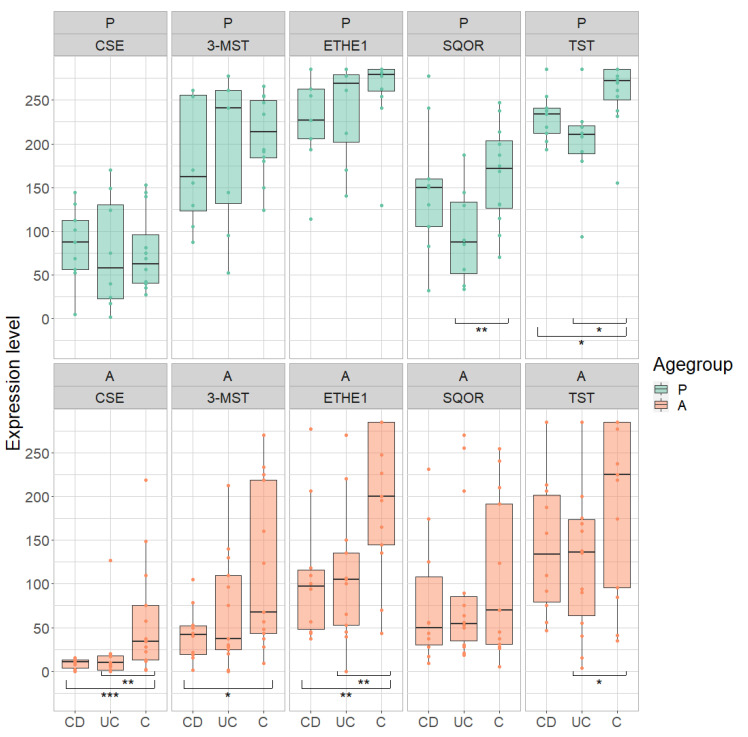
Expression of cystathionine-γ-lyase (CSE), 3-mercapto-sulfurtransferase (3-MST), ethylmalonic encephalopathy 1 protein (ETHE1), sulfide:quinone oxidoreductase (SQOR) and thiosulfate sulfurtransferase (TST) in pediatric (P) and adult (A) samples in the ascending colon in Crohn’s disease (CD), ulcerative colitis (UC) and healthy controls (C). Range of expression levels from a minimum of 0 to a maximum of 300. * = *p*-value < 0.05, ** = *p*-value < 0.01, *** = *p* < 0.001.

**Table 1 antioxidants-11-02235-t001:** Total number of patients used for this study per group and their characteristics. P-IBD: children and adolescents with inflammatory bowel disease (IBD); A-IBD: adults with IBD; pediatric controls and adult controls. CD: Crohn’s disease; UC: ulcerative colitis.

Groups		Number of Patients	Age: Mean (Range), in Years	Sex: Male/Female	Location of Biopsy		
					Terminal Ileum	Ascending Colon	Rectum
P-IBD	CD	11	12.2 (7–17)	5/6	X	X	
	UC	11	13.2 (4–16)	4/7		X	X
A-IBD	CD	11	27.8 (18–42)	3/8	X	X	
	UC	14	47.2 (19–83)	6/8		X	X
Controls	Pediatric	14	11.9 (3–17)	6/8	X	X	X
	Adults	12	57.8 (22–77)	4/8	X	X	X

**Table 2 antioxidants-11-02235-t002:** Expression as mean with standard deviation (SD), absolute reduction and percentage as well as *p*-value of the terminal ileum of the healthy control group (C) separated by age group. Pediatric (P); adult (A). cystathionine-γ-lyase (CSE), 3-mercapto-sulfurtransferase (3-MST), ethylmalonic encephalopathy 1 protein (ETHE1), sulfide:quinone oxidoreductase (SQOR) and thiosulfate sulfurtransferase (TST). ** = *p*-value < 0.01.

Controls—Terminal Ileum	3-MST	CSE	ETHE1	SQOR	TST
Expression Level, Mean (SD)					
C_P_	165 (85)	110 (53)	228 (95)	53 (37)	237 (73)
C_A_	119 (96)	42 (45)	145 (89)	38 (36)	191 (92)
Reduction C_p_ − C_A_	46 (28%)	68 (62%)	83 (36%)	15 (28%)	46 (19%)
*p*-value	0.5236	0.0086 **	0.1155	0.7397	0.3588

**Table 3 antioxidants-11-02235-t003:** Expression as mean with standard deviation (SD), absolute reduction and percentage as well as *p*-value of the rectum of the healthy control group (C) separated by age group. Pediatric (P); adult (A). cystathionine-γ-lyase (CSE), 3-mercapto-sulfurtransferase (3-MST), ethylmalonic encephalopathy 1 protein (ETHE1), sulfide:quinone oxidoreductase (SQOR) and thiosulfate sulfurtransferase (TST).

Controls—Rectum	3-MST	CSE	ETHE1	SQOR	TST
Expression Level, Mean (SD)					
C_P_	186 (47)	108 (56)	258 (33)	143 (75)	232 (70)
C_A_	118 (79)	68 (53)	216 (66)	144 (74)	211 (79)
Reduction C_p_ − C_A_	68 (37%)	40 (37%)	42 (16%)	−1 (−1%)	21 (9%)
*p*-value	0.0914	0.0852	0.072	0.8591	0.6195

**Table 4 antioxidants-11-02235-t004:** Expression as mean with standard deviation (SD), absolute reduction and percentage as well as *p*-value of the ascending colon of the healthy control group (C) separated by age group. Pediatric (P); adult (A). cystathionine-γ-lyase (CSE), 3-mercapto-sulfurtransferase (3-MST), ethylmalonic encephalopathy 1 protein (ETHE1), sulfide:quinone oxidoreductase (SQOR) and thiosulfate sulfurtransferase (TST). * = *p*-value < 0.05.

Controls—Ascending Colon	3-MST	CSE	ETHE1	SQOR	TST
Expression Level, Mean (SD)					
C_P_	210 (46)	75 (46)	262 (44)	164 (56)	258 (37)
C_A_	122 (94)	59 (64)	206 (77)	119 (92)	208 (89)
Reduction C_p_ − C_A_	88 (42%)	16 (21%)	56 (21%)	45 (27%)	50 (19%)
*p*-value	0.0224 *	0.1374	0.1259	0.275	0.2943

**Table 5 antioxidants-11-02235-t005:** Expression as mean with standard deviation (SD), absolute reduction and percentage as well as *p*-value of the terminal ileum in the Crohn’s disease group (CD) separated by age group compared to age matched controls (C). Pediatric (P); adult (A), cystathionine-γ-lyase (CSE), 3-mercapto-sulfurtransferase (3-MST), ethylmalonic encephalopathy 1 protein (ETHE1), sulfide:quinone oxidoreductase (SQOR) and thiosulfate sulfurtransferase (TST). * = *p*-value < 0.05.

CD—Terminal Ileum	3-MST	CSE	ETHE1	SQOR	TST
Expression Level, Mean (SD)					
CD_A_	65 (53)	35 (32)	127 (82)	42 (46)	126 (77)
C_A_	119 (96)	42 (45)	145 (89)	38 (36)	191 (92)
Reduction C_A_ − CD_A_	54 (46%)	7 (17%)	18 (12%)	−4 (−11%)	65 (34%)
*p*-value_A_	0.139	0.5036	0.8129	0.9343	0.2533
CD_P_	154 (88)	49 (35)	236 (75)	92 (96)	184 (99)
C_P_	165 (85)	110 (53)	228 (95)	53 (37)	237 (73)
Reduction C_P_ − CD_P_	11 (7%)	61 (56%)	−8 (−4%)	−39 (−74%)	53 (22%)
*p*-value_P_	0.8985	0.0396 *	0.8132	0.6544	0.3759

**Table 6 antioxidants-11-02235-t006:** Expression as mean with standard deviation (SD), absolute reduction and percentage as well as *p*-value of the rectum of ulcerative colitis (UC) separated by age group compared to age matched controls (C). Pediatric (P); adult (A); cystathionine-γ-lyase (CSE); 3-mercapto-sulfurtransferase (3-MST); ethylmalonic encephalopathy 1 protein (ETHE1); sulfide:quinone oxidoreductase (SQOR) and thiosulfate sulfurtransferase (TST). * = *p*-value < 0.05, **= *p*-value < 0.01, *** = *p* < 0.001.

UC—Rectum	3-MST	CSE	ETHE1	SQOR	TST
Expression Level, Mean (SD)					
UC_A_	44 (27)	6 (8)	131 (70)	72 (66)	125 (92)
C_A_	118 (79)	68 (53)	216 (66)	144 (74)	211 (79)
Reduction C_A_ − UC_A_	74 (62%)	62 (92%)	85 (39%)	72 (50%)	86 (41%)
*p*-value_A_	0.0539	3.55 × 10^−12^ ***	0.0017 **	0.021 *	0.0101 *
UC_P_	162 (51)	41 (55)	244 (48)	86 (61)	190 (80)
C_P_	186 (47)	108 (56)	258 (33)	143 (75)	232 (70)
Reduction C_p_ − UC_P_	24 (13%)	67 (62%)	14 (5%)	57 (40%)	42 (18%)
*p*-value_P_	0.238	0.0162 *	0.4812	0.0632	0.1341

**Table 7 antioxidants-11-02235-t007:** Expression as mean with standard deviation (SD), absolute reduction and percentage as well as *p*-value of the ascending colon in adults with Crohn’s disease (CD_A_) and adults with ulcerative colitis (UC_A_) compared to age matched controls (C_A_). Pediatric (P); adult (A); cystathionine-γ-lyase (CSE); 3-mercapto-sulfurtransferase (3-MST); ethylmalonic encephalopathy 1 protein (ETHE1); sulfide:quinone oxidoreductase (SQOR) and thiosulfate sulfurtransferase (TST). * = *p*-value < 0.05, ** = *p*-value < 0.01, *** = *p*-value < 0.001.

Ascending Colon	3-MST	CSE	ETHE1	SQOR	TST
Expression Level, Mean (SD)					
CD_A_	43 (31)	9 (6)	109 (78)	78 (74)	143 (79)
C_A_	122 (94)	59 (64)	206 (77)	119 (92)	208 (89)
Reduction C_A_ − CD_A_	79 (65%)	50 (85%)	97 (47%)	41 (34%)	65 (31%)
*p*-value	0.0188 *	0.0004 ***	0.0026 **	0.3113	0.0692
UC_A_	73 (66)	18 (35)	110 (76)	91 (90)	138 (87)
C_A_	122 (94)	59 (64)	206 (77)	119 (92)	208 (89)
Reduction C_A_ − UC_A_	49 (40%)	41 (70%)	96 (47%)	28 (23%)	70 (34%)
*p*-value	0.1266	0.0019 **	0.0018 **	0.6992	0.0696

**Table 8 antioxidants-11-02235-t008:** Expression as mean with standard deviation (SD), absolute reduction and percentage as well as *p*-value of the ascending colon in children with Crohn’s disease (CD_P_) and children with ulcerative colitis (UC_P_) compared to age matched controls (C_P_). Pediatric (P); adult (A); cystathionine-γ-lyase (CSE); 3-mercapto-sulfurtransferase (3-MST); ethylmalonic encephalopathy 1 protein (ETHE1); sulfide:quinone oxidoreductase (SQOR) and thiosulfate sulfurtransferase (TST). * = *p*-value < 0.05, ** = *p*-value < 0.01.

Ascending Colon	3-MST	CSE	ETHE1	SQOR	TST
Expression Level, Mean (SD)					
CD_P_	178 (72)	84 (44)	228 (54)	148 (75)	231 (28)
C_P_	210 (46)	75 (46)	262 (44)	164 (56)	258 (37)
Reduction C_P_ − CD_P_	32 (15%)	−9 (−12%)	34 (13%)	16 (10%)	27 (10%)
*p*-value	0.4635	0.5216	0.0837	0.5356	0.0308*
UC_P_	197 (87)	75 (65)	239 (57)	95 (55)	202 (54)
C_P_	210 (46)	75 (46)	262 (44)	164 (56)	258 (37)
Reduction C_p_ − UC_P_	13 (6%)	0 (0%)	23 (9%)	69 (42%)	56 (22%)
*p*-value	0.8514	0.7052	0.3238	0.0078 **	0.0118 *

## Data Availability

The data used to support the findings of this study are included within the article.

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
