# Peer review of "Hydrogen Sulfide Metabolizing Enzymes in the Intestinal Mucosa in Pediatric and Adult Inflammatory Bowel Disease"

_antioxidants, 2022, doi:10.3390/antiox11112235_

Round 1

Reviewer 1 Report

Very thorough review of the literature. The authors captured the current thinking about H2S very well.  Human biopsies very novel and difficult to get.  All results are immunohistochemistry and conclusions based on blind counting.  Some automated or digital assessment would make this stronger. 

Author Response

1. Very thorough review of the literature. The authors captured the current thinking about H2S very well.  Human biopsies very novel and difficult to get.  All results are immunohistochemistry and conclusions based on blind counting.  Some automated or digital assessment would make this stronger. 

We thank the reviewer for this comment. We agree that automatic image analysis using software is a more robust method for quantification, whereby mostly all automatic image analysis systems are based on “clear” nuclear stainings (of cell cycle) and are definitely not standardized for irregular cytoplasmatic stainings as in our presented study. Therefore, the IHC stainings were classically analyzed by evaluation of intensity and extensity in the whole biopsy by two pre-trained people blinded to the disease or age group of the analysed sample. This “pragmatical” procedure is widely accepted and was applied by our lab in many previous publications (e.g. Schneider AM, Özsoy M, Zimmermann FA, Brunner SM, Feichtinger RG, Mayr JA, Kofler B, Neureiter D, Klieser E, Aigner E, Schütz S, Stummer N, Sperl W, Weghuber D. Expression of Oxidative Phosphorylation Complexes and Mitochondrial Mass in Pediatric and Adult Inflammatory Bowel Disease. Oxid Med Cell Longev. 2022 Jan 6;2022:9151169. doi: 10.1155/2022/9151169. PMID: 35035669; PMCID: PMC8758306.)

Additionally, please refer to:

  • Rizzardi AE, Quantitative comparison of immunohistochemical staining measured by digital image analysis versus pathologist visual scoring. Diagn Pathol. 2012 Jun 20;7:42. doi: 10.1186/1746-1596-7-42. PMID: 22515559;
  • Taylor CR, Levenson RM. Quantification of immunohistochemistry--issues concerning methods, utility and semiquantitative assessment II. Histopathology. 2006 Oct;49(4):411-24. doi: 10.1111/j.1365-2559.2006.02513.x. PMID: 16978205.

Taylor CR. Quantitative in situ proteomics; a proposed pathway for quantification of immunohistochemistry at the light-microscopic level. Cell Tissue Res. 2015 Apr;360(1):109-20. doi: 10.1007/s00441-014-2089-0. Epub 2015 Jan 27. PMID: 25620411.

Reviewer 2 Report

This appears to be a good study of the expression of H2S metabolizing enzymes in the intestinal mucosa with a couple of weaknesses that probably can't be fixed easily and should not prevent the paper from getting published. The paper is well written and the arguments are supported by the data.

(1) The number of patients in each group is rather low. Therefore, the statistical correlations aren't very strong. (2) A complementary study of the gut microbiome of the same patients focusing on expression levels of H2S regulating enzymes and correlating them with the results on the intestinal mucosa would be very useful. I understand that this is a significant amount of work and is not possible to add to the present study without a long delay.

Despite these weaknesses the authors were able to show significant effects, particularly comparing expression levels in different age groups and I recommend publication with only minor revisions.

In figure captions for figures 9 and 10, please explain what (C) means (healthy controls), just like you do in figure 11.

Author Response

This appears to be a good study of the expression of H2S metabolizing enzymes in the intestinal mucosa with a couple of weaknesses that probably can't be fixed easily and should not prevent the paper from getting published. The paper is well written and the arguments are supported by the data.

(1) The number of patients in each group is rather low. Therefore, the statistical correlations aren't very strong.

We thank the reviewer for this point. Unfortunately, we cannot increase the number of patients, but we added this information in the manuscript (line 501).

 (2) A complementary study of the gut microbiome of the same patients focusing on expression levels of H2S regulating enzymes and correlating them with the results on the intestinal mucosa would be very useful. I understand that this is a significant amount of work and is not possible to add to the present study without a long delay.

This is a very important issue. The correlation of the gut microbiome would be very interesting and could definitely become a future project for our study group. The limitations this creates are mentioned in the discussion (lines 469ff and 502).

Despite these weaknesses the authors were able to show significant effects, particularly comparing expression levels in different age groups and I recommend publication with only minor revisions.

In figure captions for figures 9 and 10, please explain what (C) means (healthy controls), just like you do in figure 11.

We thank the reviewer for this comment and have changed the captions accordingly.

Round 2

Reviewer 1 Report

Thank you for the explanation about cell counting.  

Reviewer 2 Report

I am happy with the minor changes the authors made to their manuscript. Please publish as is.